# Comparison of Gut Bacterial Communities of *Locusta migratoria manilensis* (Meyen) Reared on Different Food Plants

**DOI:** 10.3390/biology11091347

**Published:** 2022-09-13

**Authors:** Qian Wang, Yusheng Liu, Xiangchu Yin

**Affiliations:** Institute of Environmental Insects, College of Plant Protection, Shandong Agricultural University, Tai’an 271000, China

**Keywords:** *Locusta migratoria manilensis*, food planst, gut bacterial community, host adaptation, 16S rRNA

## Abstract

**Simple Summary:**

Although locusts can cause major agricultural damage, they also constitute a valuable food resource. At present, *L. migratoria manilensis* has been largely domesticated, being considered an edible insect in China. Feeding variety is one of the main characteristics of *L. migratoria manilensis*. There are apparent differences in the capacity of locusts to adapt to different food plants. To elucidate the effect of different food plants (i.e., goosegrass, maize leaves, soybean leaves, and pakchoi) on the growth and development of *L. migratoria manilensis*, the gut bacterial community composition of *L. migratoria manilensis* fifth instars fed on different plants was analyzed by high-throughput sequencing. Gut bacterial communities were affected by food plants and may play an essential role in host adaption. Feeding on different food plants has significant effects on the growth and development of *L. migratoria manilensis*. The present study establishes a theoretical foundation for studying the interplay between gut bacteria structure and *L. migratoria manilensis* adaptation.

**Abstract:**

Locusts, in particular *Locusta migratoria manilensis* (Meyen), have been associated with major damages in agriculture, forestry, and animal husbandry in China. At present, *L. migratoria manilensis* has been largely domesticated, being considered an edible insect in China. Feeding variety is one of the main characteristics of *L. migratoria manilensis*. It has been demonstrated that microorganisms inhabiting the insect gut impact nutrition, development, defense, and reproduction of the insect host. The aim of the present study was to search for the adaptation mechanism of *L. migratoria manilensis* feeding on four different food plants (goosegrass, maize leaves, soybean leaves, and pakchoi) and explore changes in the gut bacterial community structure of the insect at the fifth instar nymph stage. Proteobacteria and Firmicutes were the dominant phyla, whereas *Kluyvera*, *Enterobacter*, *Pseudocitrobacter*, *Klebsiella*, *Cronobacter*, *Citrobacter*, *Lactococcus*, and *Weissella* were the dominant genera in the gut of *L. migratoria manilensis*. Principal component analysis and permutational multivariate analysis of variance (PERMANOVA) revealed significant differences in the gut microbiota structure of *L. migratoria manilensis* fed on different food plants. Moreover, functional prediction analysis revealed that metabolic and cellular processes were the most enriched categories. Within the category of metabolic processes, the most enriched pathways were carbohydrate transport and metabolism; amino acid transport and metabolism; translation, ribosomal structure, and biogenesis; cell wall/membrane/envelope biogenesis; inorganic ion transport and metabolism; and energy production and conversion. Collectively, the present results revealed that the structure of gut bacterial communities in *L. migratoria manilensis* fed on different food plants is impacted by food plants, which may play an essential part in the adaptation of the host.

## 1. Introduction

Locusts have long been known and valued in China since ancient times. Among the approximately 900 species of locusts known to date, *Locusta migratoria manilensis* (Meyen) is the leading species causing infestations in China [1]. Although locusts can cause major agricultural damage, they also constitute a valuable food resource. As a primary consumer in the food chain, locusts transform weeds, straws, and other unavailable carbohydrates and polysaccharides into assimilable protein resources, which can be used as feed alternatives for livestock and poultry farming, as well be considered a culinary specialty. As an important food resource, locusts can be raised in captivity. At present, *L. migratoria manilensis* has been largely domesticated, being considered an edible insect in China [2,3,4,5]. However, much of the costs associated with locust farming are spent on feed. Different food plants directly influence the growth and development of locusts.

Under natural conditions, plants of the two families, Gramineae and Cyperaceae, are the primary hosts of locusts. Plants of the Dicotyledon can also be temporary hosts, although are less favorable for the growth and reproduction of locusts. There are apparent differences in the capacity of locusts to adapt to different food plants [6]. Typical green forage used in the production of *L. migratoria manilensis* includes maize straw and weeds, occasionally combined with fruits, bean leaves, and vegetables. Similarly, differences in the type of forage can impact the development of *L. migratoria manilensis* [7,8,9]. It has been found that the best feed intake, weight gain, growth rate, and lowest mortality rate in *L. migratoria manilensis* are observed when feeding on Gramineae; in contrast, relatively low feed intake, weight gain, growth rate, and high mortality rate are observed when *L. migratoria manilensis* fed on lettuce leaves and carrots, which can be used as supplementary feed [7]. In addition, it has been shown that a high carrot content in the diet of farmed *L. migratoria manilensis* reportedly leads to an increase in the contents of lipids and vitamin A, whereas protein content decreases with an increase in the amount of wheat bran in the diet [10].

Insect biology and behavior must be studied as a complex ecological system of which microorganisms constitute a significant part [11,12]. The insect gut provides a particular habitat for a wide variety and number of microorganisms. The insect gut environment is affected by changes in the outer environment, and gut microecology diversity is closely related to insect species and feeding habits [13,14]. In long-term coevolution, insects and gut microorganisms have developed a symbiotic relationship [15], since the insect gut constitutes a stable environment and is a source of essential nutrients for gut microorganisms. In exchange, gut microorganisms are involved in various metabolic processes of the insect host, providing nutrients and digesting complex carbohydrates [16,17]. Studies have found that gut microorganisms can enhance host nutritional status and metabolism by synthesizing nutrients that are lacking in foods, but are essential to the host, as well as by secreting digestive enzymes that allow the host to digest certain food components. In addition, insect gut microorganisms protect the host against invasive pathogens, improve immunity, degrade exogenous biological toxins, facilitate interspecific and intraspecific communication, modulate reproduction, and enhance growth and development [18,19,20,21,22,23,24,25]. Significant differences have been described in gut microbiota composition among insects fed on different foods [26,27,28]. Hence, food is a central element affecting the composition of insect gut microbiota [29,30,31].

Therefore, the aim of the present study was to explore the effect of different food plants (i.e., goosegrass, maize leaves, soybean leaves, and pakchoi) on the growth and development of *L. migratoria manilensis*. In addition, gut bacterial community composition of *L. migratoria manilensis* fifth instars fed on different plants was analyzed by high-throughput sequencing. Furthermore, functional annotation of metagenomes was conducted to provide a basis for subsequent in-depth studies of the interplay between gut microbiota and food plants of *L. migratoria*.

## 2. Materials and Methods

### 2.1. Insect Rearing

*L. migratoria manilensis* (Meyen) were purchased from an insect breeding base in Cangzhou, Hebei, China, and reared under the following conditions: 80–100 heads per breeding cage (25 cm × 25 cm × 50 cm; L × W × H) at 28 ± 0.5 °C under 70 ± 10% relative humidity (RH) using a photoperiod (L:D) of 16:8 h and three consecutive generations. Fresh plants were inserted into water-containing sponges, and feeding and weighing were conducted daily. Insects’ developmental period, weight gain, mortality, dung production, and food intake were recorded for the calculation of each nutritional index. Newly hatched nymphs were rare in goosegrass (Gg), maize leaves (ML), soybean leaves (SL), and pakchoi (Pc) until the fifth instar stage. Experimental food plants were grown in the practical base of plant protection at Shandong Agricultural University, China. All samples for the present experiment were collected at the fifth instar nymph stage.

### 2.2. DNA Extraction

Randomly, 60 nymphs of comparable size and growth stage at the fifth instar stage were collected. After starving for two days, the surface of nymph was repeatedly washed with sterile water, then placed in 75% alcohol solution for 2 min, then rinsed with sterile water and irradiated with UV light for 3–5 min. Locust nymphs were dissected under aseptic conditions, and the whole intestine was removed; specifically, mid and hind intestines were intercepted, after which were rinsed with sterile phosphate-buffered saline (PBS) (Beijing Solarbio Science & Technology Co., Ltd., Beijing, China) [15,32], and intestinal contents of each sample were collected in sterile centrifuge tubes.

Total microbial DNA was extracted using the PowerSoil DNA Isolation Kit (MoBio, Qiagen Inc., Germantown, MD, USA) following the manufacturer’s instructions. The quality of extracted DNA was verified by horizontal gel electrophoresis in 0.8% agarose gels. Quantification of extracted DNA was determined by NanoDrop 2000 (Thermo Fisher Scientific, Waltham, MA, USA) UV-vis spectrophotometer. Each sample was conducted in three replicates.

### 2.3. Metagenomic Analysis

The universal primer pair, 341F (5′-CCTACGGGNGGCWGCAG-3′) and 805R (5′-GACTACHVGGGTATCTAATCC-3′), with a sequencing adapter at the end, were used to amplify the V3-V4 hypervariable region of the bacterial 16s rDNA gene. The PCR reaction system was composed of: PCR Phusion High-Fidelity PCR Master Mix with HF Buffer (New England Biolabs, Ipswich, MA, USA), 25 μL; DMSO, 3 μL; 3 μL of each primer; gDNA, 10 μL; nuclease-free water, q.s. to a final volume of 50 μL. PCR cycling parameters were as follows: 98 °C for 30 s; followed by 30 amplification cycles of 98 °C for 15 s, 58 °C for 15 s, and 72 °C for 15 s; followed by 1 min of final extension at 72 °C and an indefinite hold at 4 °C. PCR products were separated on 2% agarose gels and then submitted to purification.

PCR amplification of recovery products was quantified by fluorescence determination using fluorescent reagents provided within the Quant-iT PicoGreen dsDNA Assay Kit (Thermo Fisher Scientific, Waltham, MA, USA) in a FLx800 microplate reader (BioTek Instruments, Winooski, VE, USA). Based on fluorescence quantification, each sample was mixed in an appropriate proportion according to the sequencing volume required. Sequencing libraries were prepared using the TruSeq Nano DNA LT Library Prep Kit (Illumina, San Diego, CA, USA), and 2 × 300 bp double-end sequencing was performed using a MiSeq platform with the MiSeq Reagent Kit V3 (Illumina, San Diego, CA, USA) (600 cycles). Sequencing was conducted at Shandong Kaiyuan Gene Technology Co., Ltd.

### 2.4. Statistical and Bioinformatic Analysis

Firstly, quality control of raw data was conducted using Trimmomatic (v0.39) software [33]. Based on the overlap (minimum: 10 bp) between PE reads after quality control, PE reads were spliced through overlap by Flash (v1.2.11) software [34]. We used UCHIME (v4.2) [35] for the identification and removal of chimeric sequences to obtain the valid data. Based on the sequence similarity, the valid sequences were classified into multiple operational taxonomic units (OTUs) by the software VSEARCH (v2.16.0) [36] at the similarity level of 97%. All representative sequences were annotated and blasted against Silva database (v138.1, http://www.arb-silva.de (accessed on 4 January 2022)) using RDP Classifier (v2.11) [37] with a confidence threshold at 80%.

Mothur (v1.46.1) software [38] was used to calculate alpha-diversity indexes, which included ACE, Chao1, the Simpson index, and the Shannon index. In addition, principal coordinate analysis (PCoA) was used to reveal differences in gut flora composition between sample groups. Permutational multivariate analysis of variance (PERMANOVA) was performed for pairwise comparison of samples. Linear discriminant analysis (LDA) was used to identify biomarkers with statistical significance between samples with LDA scores greater than 4. Gut metagenome functions were determined by annotating pathways of OTUs against the Clusters of Orthologous Genes (COG) database using PICRUSt2 (v. 2.5.0) [39]. Differences were considered significant when *p* values were < 0.05 and highly significant when *p* values were < 0.01. SPSS 26.0 (https://www.ibm.com/products/spss-statistics (accessed on 16 March 2022)) was used for statistical analysis. The calculation formula is as follows.
$$\mathrm{AD}=\frac{100\times \left(\mathrm{Food}\;\mathrm{intake}-\mathrm{Dung}\;\mathrm{production}\right)}{\mathrm{Food}\;\mathrm{intake}}$$
$$\mathrm{ECI}=\frac{100\times \mathrm{Weight}\;\mathrm{gain}}{\mathrm{Food}\;\mathrm{intake}}$$
$$\mathrm{ECD}=\frac{100\times \mathrm{Weight}\;\mathrm{gain}}{\mathrm{Food}\;\mathrm{intake}-\mathrm{Dung}\;\mathrm{production}}$$
$$\mathrm{Growth}\;\mathrm{rate}=\frac{\mathrm{Weight}\;\mathrm{gain}\left(\mathrm{mg}\right)}{\mathrm{Developmental}\;\mathrm{time}\left(\mathrm{days}\right)}$$
Overall vitality=Survival rate×Growth rate

## 3. Results

### 3.1. Development Rate of L. migratoria manilensis Reared on Different Food Plant

*L. migratoria manilensis* had the highest food utilization rate, weight gain, and the lowest mortality rate when grown on maize leaves (ML), followed by goosegrass (Gg), soybean leaves (SL), and pakchoi (Pc) (Table 1, Figure 1). In contrast, the average generation period was intermediate for *L. migratoria manilensis* fed on SL; the nymph stage was the longest, and the adult stage was the shortest in *L. migratoria manilensis* fed on Pc. All four plants species completed their life cycles (Table 2). The growth rate and overall vitality of *L. migratoria manilensis* fed on ML were significantly higher than those fed on the other food plants, with the shortest developmental period. The growth rate and overall vitality of *L. migratoria manilensis* fed on Pc were substantially lower than those fed on the other food plants, with the longest developmental period (Figure 2).

### 3.2. Metagenomic Analysis

Four groups of 12 samples were sequenced. A total of 1,265,960 reads were obtained. After quality control, a total of 1,236,884 effective reads remained (Appendix A). Cluster analysis resulted in 2838 OTUs, which corresponded to 26 phyla, 55 classes, 111 orders, 155 families, 247 genera, and 387 species. The sample scarcity curve (Appendix A) and the Shannon diversity index scarcity curve (Appendix A) indicate that sequencing volume was sufficient, the sequencing depth was saturated, and increasing the sample volume would not produce more OTUs. Moreover, Good’s coverage index was used to verify sequencing completeness. The results showed that sequencing coverage was greater than 99%, showing that most microbial species found in the samples were characterized.

### 3.3. Metagenomic Analysis

For alpha-diversity analysis, ACE and Chao1 indexes reflect the richness of the microbial community in a sample; the larger the value of these indexes, the higher the community richness. The gut microbiota in *L. migratoria manilensis* fed on SL had a lower abundance compared to the other treatment groups. In addition, Shannon and Simpson indexes reflect diversity in the gut microbial community; a higher diversity in a sample is indicated by higher Shannon index values and lower Simpson index values. No significant difference was found among the samples across the four feeding conditions. Thus, these results showed that the gut microbiota of *L. migratoria manilensis* reared on four different food plants had high species diversity and richness, although no significant difference was found among the samples (Figure 3).

At the phylum level, the top ten phyla in relative abundance were Proteobacteria, Firmicutes, Bacteroidota, Actinobacteriota, Acidobacteriota, Verrucomicrobiota, Planctomycetota, Chloroflexi, Gemmatimonadota, and Myxococcota (Figure 4A). Among them, Proteobacteria was the dominant phylum, showing the highest relative abundance in *L. migratoria manilensis* fed on Pc (94.66 ± 4.511%); the relative abundance of Proteobacteria in the other feeding groups was as follows: 77.18 ± 9.762% in *L. migratoria manilensis* fed on ML; 87.86 ± 6.509% on Gg; and 45.19 ± 9.619% on SL. In contrast, Firmicutes was the dominant phylum in the gut microbiota of *L. migratoria manilensis* fed on SL, whose relative abundance was 54.80 ± 9.63% (Appendix A).

At the family level, the top ten families in relative abundance were Enterobacteriaceae, Streptococcaceae, Lactobacillaceae, Morganellaceae, Hafniaceae, Erwiniaceae, Enterococcaceae, Moraxellaceae, Yersiniaceae, and Pseudomonadaceae (Figure 4B). The highest relative abundances of Enterobacteriaceae were observed in the gut microbiota of *L. migratoria manilensis* fed on ML, Gg, and Pc (75.88 ± 9.106%, 85.92 ± 7.005%, and 84.37 ± 5.735%, respectively). In contrast, the relative abundance of Enterobacteriaceae in *L. migratoria manilensis* fed on SL was 44.79 ± 9.69%. The relative abundance of Streptococcaceae and Lactobacillaceae was significantly higher in the gut microbiota of *L. migratoria manilensis* fed on SL (35.56 ± 9.11% and 18.85 ± 1.28%, respectively) (Appendix A).

For beta-diversity analysis, a clustering heat map was created to reveal the dynamics of *L. migratoria manilensis* gut microbiota fed on different food plants, based on the top 20 relative abundances of bacteria at the genus level. Collectively, the gut microbiota of *L. migratoria manilensis* fed on ML, Gg, and Pc was similar in composition at the genus level, in which *Kluyvera*, *Enterobacter*, and *Pseudocitrobacter* were the dominant genera. In contrast, *Lactococcus* and *Weissella* were the dominant genera in the gut microbiota of *L. migratoria manilensis* fed on SL, with *Lactococcus* present in significantly higher relative abundance compared to the other groups (Figure 5).

In order to identify biomarkers of bacteria and different levels of taxa change that could enable distinguishing the different sample groups, LDA effect size (LEfSe) was used on OTUs at various taxonomic ranks (kingdom, phylum, class, order, family, genus, and species) with standard LDA values > 4 (Appendix A). In addition, a cladogram at taxonomic ranks phylum, class, order, family, genus, and species was used to elucidate the distribution of changes at various taxonomic ranks (Figure 6).

The gut bacterial community of *L. migratoria manilensis* fed on different food plants was composed of different taxa (LDA > 4), mainly Proteobacteria and Firmicutes. There were 11 different taxa in SL, primarily Firmicutes; ten groups were found in Pc, mainly Proteobacteria; one different group was found in ML, which corresponded to Firmicutes; one different group was found in Gg, which corresponded to Proteobacteria. Collectively, these results indicated that host plant had a significant effect on shaping the gut microbiota structure of *L. migratoria manilensis*.

### 3.4. Functional Annotation of Gut Microbial Community in L. migratoria manilensis

To gain insights into the role of gut microbiota in *L. migratoria manilensis*, the PICRUSt2 software was used to predict the function of obtained metagenomes by annotating against the COG database.

Overall, the results showed that most functional prediction categories are related to metabolic and cellular processes. The main metabolic functions include carbohydrate transport and metabolism; amino acid transport and metabolism; translation, ribosomal structure and biogenesis; cell wall/membrane/envelope biogenesis; inorganic ion transport and metabolism; energy production and conversion; transcription, thus, representing the most active functions in the gut microbiota of *L. migratoria manilensis* fed on four different food plants (Figure 7). The functional category translation, ribosomal structure, and biogenesis were significantly more enriched in SL compared to the other treatment groups, whereas differences between the remainder of the samples were not significant (Figure 8D). Moreover, carbohydrate transport and metabolism were found to be enriched the lowest in Pc, and the highest in Gg, whereas differences in the proportion of OTUs mapped to this category were not significant between the other two sample groups (Figure 8B). Considering amino acid transport and metabolism, the lowest enrichment was found in SL, whereas no difference was found among the other three sample groups (Figure 8A). Finally, the proportion of OTUs mapped to nucleotide transport and metabolism category was comparable to those mapped to translation, ribosomal structure, and biogenesis category (Figure 8C).

## 4. Discussion

As the most widely distributed locust species in China, *L. migratoria manilensis* (Meyen) has a vigorous life cycle, with two to four generations yearly, and can survive under various environmental conditions. In addition, its feeding, migration, and aggregation habits are versatile. It is a typical polyphagous insect that feeds on plants of the families Gramineae and Cyperaceae, as well as occasionally on certain plants of Dicotyledon [40,41,42]. The present study aimed to explore the differences in the gut bacterial community of *L. migratoria manilensis* fed on four different food plants. *L. migratoria manilensis* fed on Gramineae showed better growth performance than those fed on Dicotyledon. Significant differences were found in the adaptability of *L. migratoria manilensis* to different food plant, mainly in terms of growth parameters and differences in food usage. Taken together, the results described herein provide a more integrative understanding of the connections between *L. migratoria manilensis* and its symbiotic gut microorganisms. Based on the data, it is tentatively concluded that food plants affected the diversity and abundance of the gut bacterial community of *L. migratoria manilensis*, which revealed a sophisticated relationship between the gut bacteria of *L. migratoria manilensis* and food plants, thus providing a theoretical basis for comprehending the adaptation mechanisms of *L. migratoria manilensis* to food plants.

Previous studies have shown that insect gut microbial composition is significantly influenced by host species and diet [32,43]. However, the influence of the host dietary is greater compared to the influence of host species [44,45,46]. The growth rate, fecundity, and survival rate of locusts fed on different food plants differ widely [6,40,47], which will affect gut microorganisms development. Moreover, it has been shown that food plants induce differences in the gut microenvironment, and the diversity in the abundance of gut microbes can change [48]. Therefore, the abundance of gut bacteria depends on the composition and content of food plant upon which locusts feed. Thus, changes in food plant will necessarily lead to changes in the contents of nutrients available to gut bacteria, hence resulting in variations in the gut microbiota composition [49,50,51]. The diversity of gut bacteria in *L. migratoria manilensis* changed to varying degrees after feeding on four different food plants, and the gut bacterial community changed in response to changes in the dietary. Similar changes were observed in the diversity and structure of gut bacterial communities in *Grapholita molesta* and *Plutella xylostella* fed on different food plants [15,52].

Differences in the relative abundance of phyla enable a comprehensive assessment of differences in the composition of the gut bacterial community, as each phylum usually contributes to different functions. Proteobacteria and Firmicutes were the dominant bacterial phyla in the four treatment groups. Many studies have shown that Proteobacteria and Firmicutes predominate in many insects’ gut microbiota [18,53,54,55,56], playing an essential role in carbohydrate metabolism, amino acid metabolism, energy production, and membrane transport [57,58,59]. Moreover, it has been described that colonization of insect gut to form a stable microbiota may be the result of long-term interaction and adaptation between host insects and food plants, which is essential for insects to adapt to feeding on specific food plants [52]. In the present study, the gut bacterial community of *L. migratoria manilensis* was closely associated with the host plant, which may play an essential part in food ingestion and nutrient utilization. Future research combining metabolomics would elucidate how gut bacterial communities affect insects’ metabolic processes.

Herein, Enterobacteriaceae, Streptococcaceae, and Lactobacillaceae were found to be the predominant families in the gut bacterial community of *L. migratoria manilensis* in the four treatment groups; in particular, Enterobacteriaceae was the most predominant in all samples. It has been reported that Enterobacteriaceae is commonly found in the gut microbiota of many insects [18,58]. Previous studies have shown that Enterobacteriaceae plays an essential role in the host’s glucose metabolism, digestion, as well as courtship and reproductive behaviors [32,60,61]. In addition, the high abundance of Enterobacteriaceae may be related to host adaptability [15,32]. In the present study, functional prediction analysis revealed that Enterobacteriaceae is related to carbohydrate transport and metabolism. Within the family Enterobacteriaceae, the most abundant genus was *Kluyvera*, a Gram-negative bacterium with cellulose-degrading and nitrogen fixation abilities that can improve the host’s nutrition by providing nitrogenous compounds necessary for amino acid synthesis [62,63]. Plants of the family Gramineae contain high amounts of cellulose, which may account for the higher abundance of *Kluyvera* in the samples. In the family Streptococcaceae, the most abundant genus was *Lactococcus* in SL. It has been reported that *Lactococcus* can be found in the viscera of different animals (including insects), which may be responsible for lignocellulose digestion and fermentation [20,64]. Combined with functional prediction analysis, *Lactococcus* could be related to the category translation, ribosomal structure, and biogenesis. The high protein content in SL may account for the high abundance of *Lactococcus* in this sample.

In addition, the five genera with the highest relative abundance in the gut bacterial community of *L. migratoria manilensis* fed on Gg and ML were highly comparable, which included *Kluyvera*, *Enterobacter*, *Pseudocitrobacter*, *Lactococcus*, and *Weissella*. However, the abundance of these genera varied. The top five genera in terms of relative abundance in the gut bacterial community in *L. migratoria manilensis* fed on Pc were *Kluyvera*, *Enterobacter*, *Pseudocitrobacter*, *Cronobacter*, and *Lactococcus*. In contrast, the top five genera in relative abundance in the gut bacterial community in *L. migratoria manilensis* fed on SL were *Lactococcus*, *Weissella*, *Enterobacter*, *Pseudocitrobacter*, and *Klebsiella*. PERMANOVA analyses revealed substantial differences between treatment groups (R^2^ = 0.601; *p* = 0.002). Thus, these results indicate that the abundance of the gut microbiota varied according to the host plant source, even those belonging to the family Gramineae. It has been previously stated that the gut microbiota structure changes considerably according to the host requirements for growth and development [52,65].

Plants of the families Gramineae and Cyperaceae are the primary hosts of locusts in nature, whereas plants of Dicotyledon can also serve as temporary host. Feeding on different food plant had a significant impact on the growth and reproduction of *L. migratoria manilensis* [6], as well as on species diversity and abundance in the insect gut bacterial community [66]. Thus, it can be stated that there were obvious differences in adaptation to different food plants in *L. migratoria manilensis*, and plants of the family Gramineae are the most suitable food plant for locusts. However, previous studies have been conducted to identify some gut bacteria by isolation and culture, and have only involved gut microbiota that feeds on plants of the family Gramineae. In the present study, it was shown that the structure of the gut bacterial community of *L. migratoria manilensis* fed on SL and Pc differed significantly from that of locusts fed on Gg and ML. Moreover, the abundance and diversity of the gut microbiota in SL and Pc treatment groups were not lower than those in Gg and ML treatment groups. Combined with COG functional prediction analysis, it can be stated that the gut microbiota structure of *L. migratoria manilensis* changed qualitatively and quantitatively in response to the characteristics of food plants in order to adapt to the type of feed source as well as to meet the nutritional requirements of the host.

Collectively, the results discussed in the present study indicated that host plant could affect the gut bacterial structure of *L. migratoria manilensis*. Gut bacterial communities were affected by the host’s diet and may play an essential role in host adaption. However, gut bacterial structures were measured after three rearing generations in the present study, whereas it would be ideally to explore differences in gut bacterial structure in each generation. Such information would contribute to a higher comprehension of the dynamic changes in the gut bacteria of *L. migratoria manilensis* during adaptation to various food plants over different generations. A study combining multi-omics techniques to elucidate the vital role of gut bacteria in host adaptation would enable finding new targets for controlling locusts. Finally, the present study establishes a theoretical foundation for studying the interplay between gut bacteria structure and *L. migratoria manilensis* adaptation.

## 5. Conclusions

In the present work, we explored the effect of different food plants (i.e., goosegrass, maize leaves, soybean leaves, and pakchoi) on the growth and development of *L. migratoria manilensis*. In addition, changes in the gut bacterial community structure of the insect at the fifth instar nymph stage were explored by 16S rDNA sequencing. *L. migratoria manilensis* that fed on Gramineae showed better growth performance than those that fed on Dicotyledon. Collectively, the host plant affected the diversity and abundance of the gut bacterial community of *L. migratoria manilensis*. The gut microbiota structure of *L. migratoria manilensis* changed qualitatively and quantitatively in response to the characteristics of food plants in order to adapt to the type of feed, as well as to meet the nutritional requirements of the host. The current study provides a theoretical basis for a better understanding of the adaptation mechanisms of *L. migratoria manilensis* to its food plant, as well as sheds new light on the role of gut bacteria in host adaptation and nutrition.

## Figures and Tables

**Figure 1 biology-11-01347-f001:**
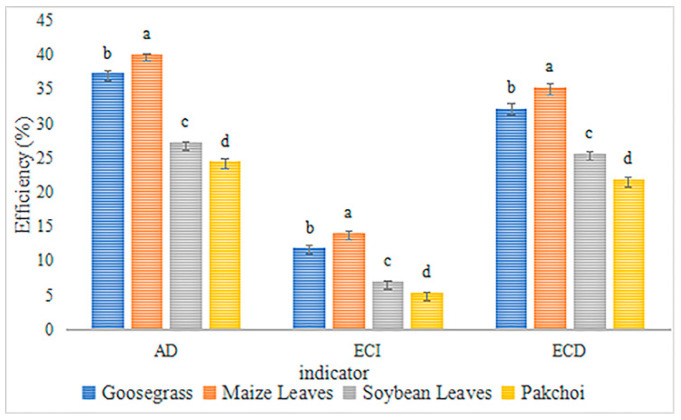
Food utilization efficiency by *Locusta migratoria manilensis* (Meyen) reared on four food plants. AD, ECI, and ECD indicated approximate digestibility (means ± SE), efficiency of conversion of ingested food (mean ± SE) and efficiency of conversion of digested food (mean ± SE), respectively. Significant differences within AD, ECl, and ECD are indicated by lowercase letters (ANOVA, *p* < 0.05), respectively.

**Figure 2 biology-11-01347-f002:**
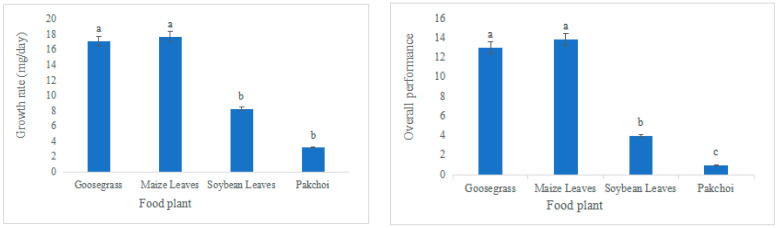
Growth rate and overall performance of *Locusta migratoria manilensis* reared on four food plants. Bars marked by different lowercase letters are significantly different based on Turkey’s HSD analysis at *p* < 0.05.

**Figure 3 biology-11-01347-f003:**
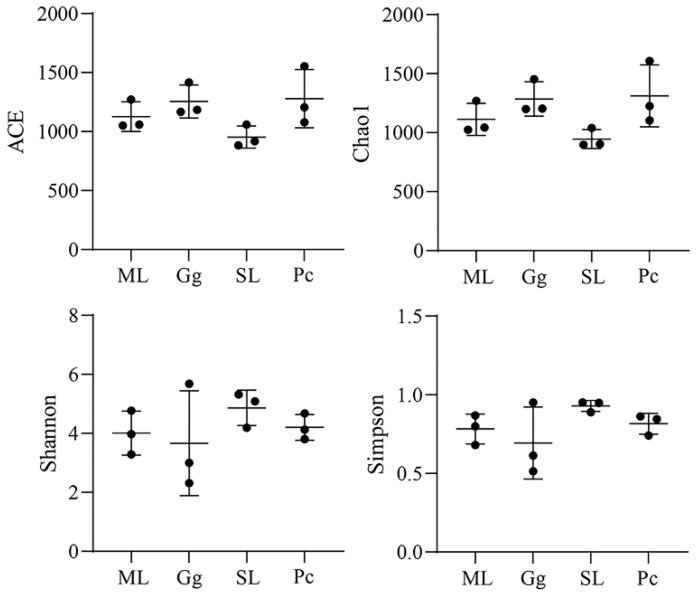
Box plots of ACE, Chao 1, and Shannon and Simpson indexes of gut microbial communities in *Locusta migratoria manilensis* fed on four different food plants. ML: maize leaves; Gg: goosegrass; SL: soybean leaves; Pc: pakchoi.

**Figure 4 biology-11-01347-f004:**
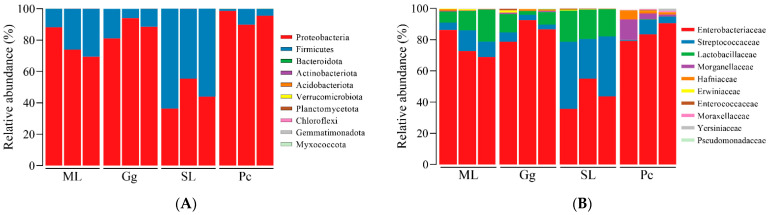
Bacterial composition of the top 10 relative abundances of bacteria at the phylum (**A**) and (**B**) family levels in the gut of *Locusta migratoria manilensis* fed on four different food plants. ML: maize leaves; Gg: goosegrass; SL: soybean leaves; Pc: pakchoi. Each color represents a phylum in (**A**) and a family in (**B**), and the height of the color block represents the proportion of relative abundance of that species.

**Figure 5 biology-11-01347-f005:**
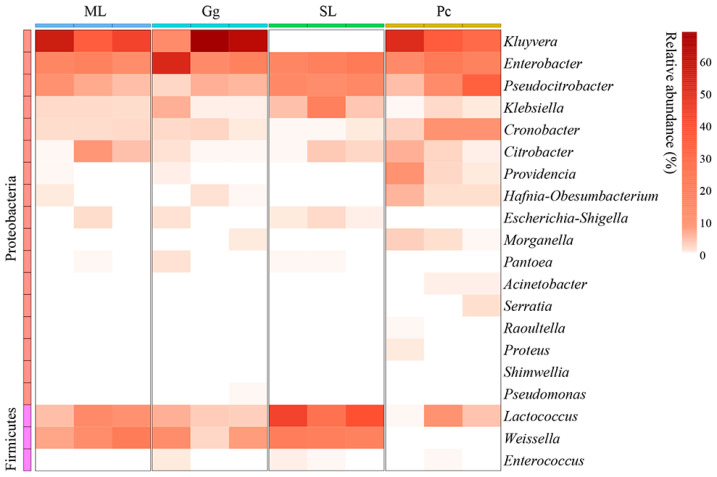
Clustering heat map of the 20 most abundant genera in the gut bacterial community of *Locusta migratoria manilensis* fed on four different food plants. ML: maize leaves; Gg: goosegrass; SL: soybean leaves; Pc: pakchoi.

**Figure 6 biology-11-01347-f006:**
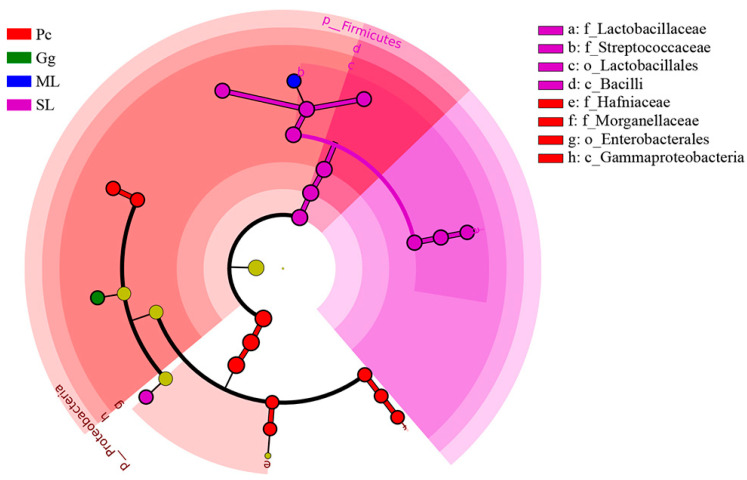
Cladogram of bacterial biomarkers from the phylum to genus (innermost ring to outermost ring) level, with LDA score > 4. ML: maize leaves; Gg: goosegrass; SL: soybean leaves; Pc: pakchoi. Differential bacterial taxa are marked by lowercase letters. The circles at different taxonomic levels represent a taxon at that level, and the diameter of the circle corresponds to the relative abundance. Different colors represent different groups species, with yellow representing species with insignificant differences and the other different species as the group with the highest abundance of the species.

**Figure 7 biology-11-01347-f007:**
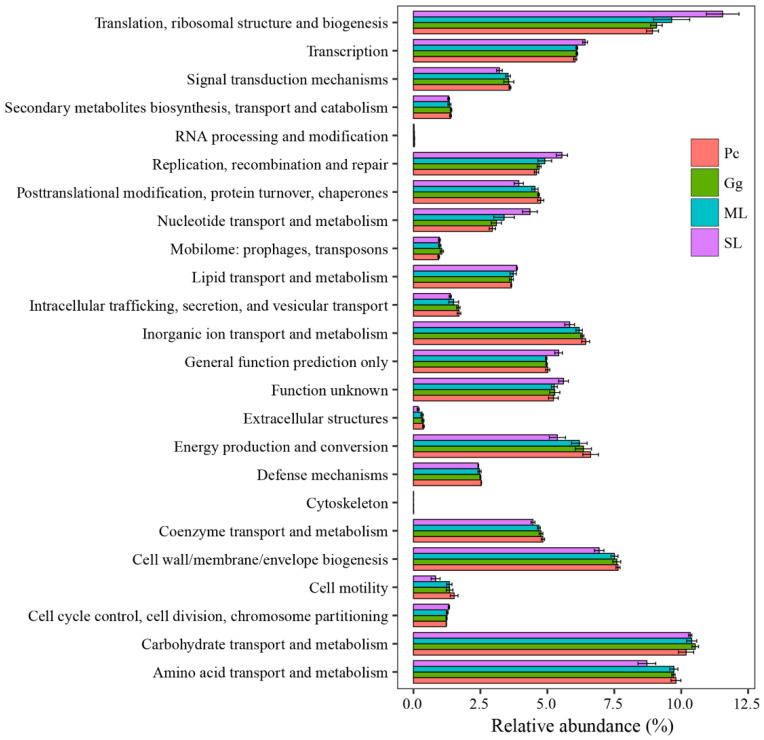
Comparison of predicted Clusters of Orthologous Genes (COG) functional categories of the gut bacterial community in *Locusta migratoria manilensis* fed on different food plants. ML: maize leaves; Gg: goosegrass; SL: soybean leaves; Pc: pakchoi.

**Figure 8 biology-11-01347-f008:**
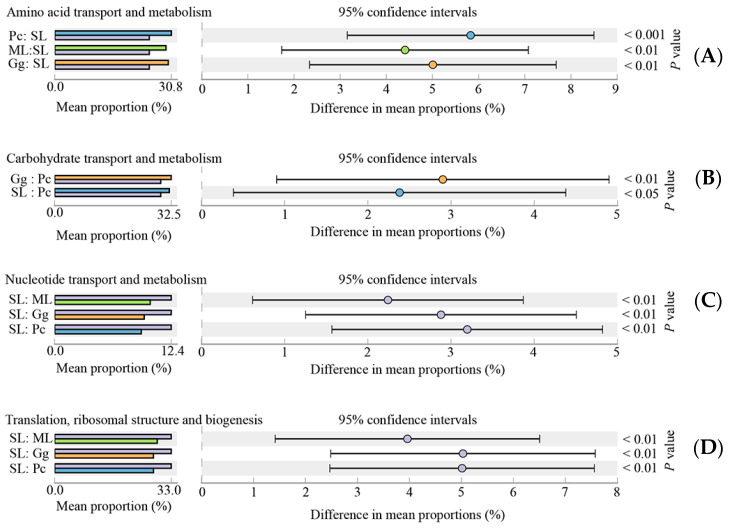
Significant differences (*p* < 0.05) in the abundance of operational taxonomic units (OTUs) mapped to predicted Clusters of Orthologous Genes (COG) functional categories amino acid transport and metabolism (**A**), carbohydrate transport and metabolism (**B**), nucleotide transport and metabolism (**C**), and translation, ribosomal structure, and biogenesis (**D**). ML: maize leaves; Gg: goosegrass; SL: soybean leaves; Pc: pakchoi.

**Table 1 biology-11-01347-t001:** Productive parameters of the nymph and Egg production of the female of *Locusta migratoria manilensis* reared on four food plants.

Evaluated Parameters	Goosegrass	Maize Leaves	Soybean Leaves	Pakchoi
Food intake (g)	2.715 ± 0.028 ^a^	2.752 ± 0.029 ^a^	1.963 ± 0.022 ^b^	1.448 ± 0.013 ^c^
Dung production (g)	1.625 ± 0.015 ^a^	1.643 ± 0.015 ^a^	1.214 ± 0.014 ^b^	0.973 ± 0.044 ^c^
Weight gain (g)	0.385 ± 0.008 ^a^	0.392 ± 0.005 ^a^	0.244 ± 0.004 ^b^	0.110 ± 0.001 ^c^
Survival rate (%)	76.40 ± 0.51 ^a^	78.20 ± 0.58 ^a^	47.60 ± 0.93 ^b^	27.60 ± 1.36 ^c^
Egg production (eggs)	251.12 ± 12.72 ^a^	273.35 ± 13.47 ^a^	85.52 ± 6.45 ^b^	84.97 ± 11.04 ^b^

Data are reported as the mean ± SE. Different lowercase letters (a, b, c) indicate significant differences in the mean values of relative abundance within the same line. (One-way ANOVA, Tukey post-hoc test, *p* < 0.05).

**Table 2 biology-11-01347-t002:** Developmental durations of *L. migratoria manilensis* reared on four food plants.

Developmental Stage		Goosegrass	Maize Leaves	Soybean Leaves	Pakchoi
Pre-adult stage (d)		43.68 ± 0.32 ^c^	43 ± 0.29 ^c^	51.63 ± 0.4 ^b^	55.43 ± 0.41 ^a^
Adult pre-oviposition period (d)		14.72 ± 0.33 ^b^	14.26 ± 0.27 ^b^	16.43 ± 0.34 ^a^	16.13 ± 0.4 ^a^
Adult stage (d)	Female insects	43.18 ± 1.7 ^a^	44.35 ± 1.49 ^a^	22.29 ± 0.77 ^b^	21.38 ± 0.75 ^b^
	Male insects	41.5 ± 3.14 ^a^	42.13 ± 2.91 ^a^	19.4 ± 1.21 ^b^	19.67 ± 1.2 ^b^
Average generation period (d)		66.67 ± 0.43 ^c^	65.53 ± 0.51 ^d^	70.31 ± 0.45 ^b^	74.24 ± 0.41 ^a^

Data are reported as the mean ± SE. Different lowercase letters (a, b, c) indicate significant differences in the mean values of relative abundance within the same line. (One-way ANOVA, Tukey post-hoc test, *p* < 0.05).

## Data Availability

The original 16S rRNA sequence data are available in the NCBI Sequence Read Archive under accession number PRJNA858491.

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
