# Peer review of "Comparison of Gut Bacterial Communities of Locusta migratoria manilensis (Meyen) Reared on Different Food Plants"

_biology, 2022, doi:10.3390/biology11091347_

Round 1

Reviewer 1 Report

The manuscript is presented in a very professional way with only few minor mistakes of English. Otherwise, it is fine for me.

Author Response

Dear Reviewer:

Thank you for your approval on our manuscript entitled "Food sources drive development and variation in Locusta migratoria manilensis (Meyen) gut bacterial community" (ID: biology-1882540) Acknowledgement. Your affirmation is of great importance to our research. We have partially revised the article to make it more clearly organized and we hope to receive your approval. These changes will not influence the content and frameword of paper. And here we did not list the changes but markes in red revised paper.

Once again, thank you very much for your affirmation!

Reviewer 2 Report

The work study the effect of different host plants on the growth and development of L. migratoria manilensis, and the effect on the composition of gut bacterial community. The result showed that the communities and their expressed genes of gut bacterial in L. migratoria manilensis were affected by host plant, and gut bacterial might play an essential role in host food adaption. Such result enriched the content of insect gut bacterials and its biological function on host, which have some scientific and application value. But there are some points need to be revised.

1.       The title “Food sources drive development and variation in Locusta migratoria manilensis (Meyen) gut bacterial community” set the “Food sources” as the head-word, Instead of the gut bacterial. The article mainly investigate the variation in community and gene expression of gut bacteria when feeding on different food plant. The core is the gut bacteria. The present title did not well reflect the main content of the article. Moreover, the “variation” in the title should more specific. Please revise it. 

2.       “gut microbiota structure of L. migratoria manilensis changed qualitatively and quantitatively in response to the characteristics of host plants in order to adapt to the type of feed source as well as to meet the nutritional requirements of the host”. The whole article should take the gut microbiota as the leading topic, and were organized like this expression.

3.       It already knew that feeding on different host plants had a significant impact on the growth and reproduction of L. migratoria manilensis [6], as well as on species diversity and abundance in the insect gut bacterial community [66]. Thereby, what’s the difference of this work to these previous reported works, it’s the repeat of previous works? What’s the meaning of this study? Reorganize the introduction part and give the scientific reason for carrying out the present research please.

4.       The expression of “Food source” is not precision, it could be misunderstanding as the same food plant from different places. But it was not the meaning in this article. Please use a more precise words such as “food plant”. Many other similar usage such as Food sources, dietary sources, dietary species, host species and host plants were used in the article too. Please reconsider these usage and select a correct and unambiguous one.

5.       Line 233-235, the sentence is wrong expressed obviously, revise the expression

6.       Line 164 and 314: L. migratoria manilensis, The species name should be in italics

7.       Line 376: “….which will affect gut development”, no content involve the gut development in the article.

8.       Line 453-455. “However, gut bacterial structures were measured after three rearing generations in the present study, whereas it would be ideally to explore differences in gut bacterial structure in each generation”, this work involved the three rearing generations?

Author Response

Dear Reviewer:

Thank you for your comments concerning our manuscript entitled “Food sources drive development and variation in Locusta migratoria manilensis (Meyen) gut bacterial community” (ID: biology-1882540). Those comments are all valuable and very helpful for revising and improving our paper, as well as the important guiding significance to our research. We have studied comments carefully and made corrections, which we hope will be approved by you. Revised portions are marked in red on the paper. The main corrections in the paper and responses to your comments are as follows:

  1. The title “Food sources drive development and variation in Locusta migratoria manilensis (Meyen) gut bacterial community” set the “Food sources” as the headword, Instead of the gut bacterial. The article mainly investigate the variation in community and gene expression of gut bacteria when feeding on different food plant. The core is the gut bacteria. The present title did not well reflect the main content of the article. Moreover, the “variation” in the title should be more specific. Please revise it.

Response: Thank you for your valuable suggestions. We have made changes to address this issue. The original title really did not express the theme of the manuscript very clearly, so we have changed the title to “Comparison of Gut Bacterial Communities of Locusta migratoria manilensis (Meyen) Reared on Different Food Plant”, which is simpler and more precise.

  1. “Gut microbiota structure of migratoria manilensis changed qualitatively and quantitatively in response to the characteristics of host plants in order to adapt to the type of feed source as well as to meet the nutritional requirements of the host”. The whole article should take the gut microbiota as the leading topic, and were organized like this expression.

Response: We are incredibly grateful to you for pointing out this problem. We have revised and removed as much content as possible that does not reflect the theme of the manuscript.

  1. It already knew that feeding on different host plants had a significant impact on the growth and reproduction of migratoria manilensis [6], as well as on species diversity and abundance in the insect gut bacterial community [66]. Thereby, what’s the difference of this work to these previous reported works, it’s the repeat of previous works? What’s the meaning of this study? Reorganize the introduction part and give the scientific reason for carrying out the present research please.

Response: Thank you for your question, this part is indeed an oversight in our writing and not rigorous enough. We have explained this issue in lines 444-447 of the manuscript.

  1. The expression of “Food source” is not precision, it could be misunderstanding as the same food plant from different places. But it was not the meaning in this article. Please use a more precise words such as “food plant”. Many other similar usage such as Food sources, dietary sources, dietary species, host species and host plants were used in the article too. Please reconsider these usage and select a correct and unambiguous one.

Response: Thank you for your suggestion, we have revised these words in the manuscript that were not expressed accurately enough and have taken your advice and replaced the words with "Food plant".

  1. Line 233-235, the sentence is wrong expressed obviously, revise the expression.

Response: Thank you for the important feedback, we have made adjustments to this wrong expressed sentence. Revised portions are marked in red on the paper.

  1. Line 164 and 314: migratoria manilensis, The species name should be in italics

Response: Thank you for your questions about the details, we have made changes to those details.

  1. Line 376: “….which will affect gut development”, no content involve the gut development in the article.

Response: Thank you for your question about the details, it was a mistake in my expression and we have made a change to the sentence.

  1. Line 453-455. “However, gut bacterial structures were measured after three rearing generations in the present study, whereas it would be ideally to explore differences in gut bacterial structure in each generation”, this work involved the three rearing generations?

Response: Thank you for your question, and I would like to explain it to you. For this work, we did keep the insects continuously for three generations. Due to the high mortality rate of the first generation, the survival of the insects was unstable, and the experiment was started only after the survival rate was stable for three generations of continuous indoor rearing. In addition, according to the design of previous experiments, most of them were reared for three generations before conducting various experiments.

We appreciate for  Reviewers’ warm work earnestly and hope that correction will meet with approval.

Once again, special thanks to you for your good comments.

Reviewer 3 Report

Wang et al characterize the bacterial community associated with Locusta migratoria manilensis reared on four different diets. The authors also monitored the impact of different diets on L. migratoria manilensis development. The authors use their data to predict what food crops will be best suited for rearing Locusta migratoria manilensis efficiently. Here are a few comments to make the manuscript better.

My main concern about the manuscript is that it wasn’t exactly clear to me what the main focus of the manuscript is. Are the authors primarily interested in knowing how diet influences gut microbial composition in L. migratoria manilensis or primarily interested in the types of gut microbe that facilitate the most ‘efficient’ growth of L. migratoria manilensis. From the way the manuscript is currently written it is hard for the reader to discern what the main goal of the manuscript is.

Methods: The authors do not describe in enough detail how data was collected for this study. For instance, the authors do not describe how feeding was performed, how food intake was measured or how dung production was estimated. These are just a few examples. In general, not enough information is provided by the authors for the experiments used in this study to be evaluated and/or replicated.

Results: Some of the data is overinterpreted in some places. For instance, at Lines 213 – 282, the authors indicate that the changes seen in gut microbial communities is driven by the diet of the of L. migratoria manilensis.  Without a reference point or appropriate controls (i.e characterizing the microbial composition pre-feeding on different diet) the authors do not have enough data to come to this conclusion. Also at, Lines 306 – 326. No new info is provided. Almost all major metabolic pathways are shown as important and the relative abundances of the genes somewhat corresponds to their relative abundance in the genomes of most microorganisms.

Figures: The manuscript currently has 11 figures. I suggest the authors move half of the figures to supplemental and focus on the core figures that best enhance comprehension of their work. Also, key details about how calculations were performed to generate certain figures is missing. For example, how did the authors calculate food utilization efficiency and growth rate etc (Figs 1 & 2)? Table 2 and Figure 3 can be moved to supplemental as an example.

Discussion: As mentioned in the results some of the info here is over-interpreting the data the authors collected. For instance, at Lines 357 to 363 the authors talk about ‘revealing sophisticated symbiotic relationships between gut bacteria and L. migratoria manilensis. The authors do not have any data to support this statement.

Author Response

Dear Reviewer:

Thank you for your comments concerning our manuscript entitled “Food sources drive development and variation in Locusta migratoria manilensis (Meyen) gut bacterial community” (ID: biology-1882540). Those comments are all valuable and very helpful for revising and improving our paper, as well as the important guiding significance to our research. We have studied comments carefully and made corrections, which we hope will be approved by you. Revised portions are marked in red on the paper. The main corrections in the paper and responses to the reviewer’s comments are as follows:

  1. My main concern about the manuscript is that it wasn’t exactly clear to me what the main focus of the manuscript is. Are the authors primarily interested in knowing how diet influences gut microbial composition in L. migratoria manilensis or primarily interested in the types of gut microbe that facilitate the most ‘efficient’ growth of L. migratoria manilensis. From the way the manuscript is currently written it is hard for the reader to discern what the main goal of the manuscript is.

Response: Thank you for your question about this manuscript, and to explain it here, the main point of the manuscript is “Gut microbiota structure of L. migratoria manilensis changed qualitatively and quantitatively in response to the characteristics of host plants in order to adapt to the type of feed source as well as to meet the nutritional requirements of the host”. We have revised and removed as much content as possible that does not reflect the theme of the manuscript.

  1. Methods: The authors do not describe in enough detail how data was collected for this study. For instance, the authors do not describe how feeding was performed, how food intake was measured or how dung production was estimated. These are just a few examples. In general, not enough information is provided by the authors for the experiments used in this study to be evaluated and/or replicated.

Response: Thank you for the important feedback. I will explain here in response to these questions. In the method section, we are writing according to the way materials and methods are written in most papers. We have added a part of the manuscript, and because of the tedious content, the specific feeding process is not written, so I will elaborate here for you: feed the insects every morning and evening at 8:00 a.m. Weigh two rations of fresh plants, one inserted into a sponge containing water to feed the insects to ensure the water content of the plants, and the other Dried and weighed. The remaining plants in the cage were removed and dried and weighed, and food intake was calculated. The dung in the cages was collected and dried and weighed to calculate the dung production. Check the death of insects in the cages daily and replenish them with the same size insects in time and measure the weight of insects every three days to calculate the weight gain.

  1. Results: Some of the data is overinterpreted in some places. For instance, at Lines 213 – 282, the authors indicate that the changes seen in gut microbial communities is driven by the diet of the of L. migratoria manilensis. Without a reference point or appropriate controls (i.e characterizing the microbial composition pre-feeding on different diet) the authors do not have enough data to come to this conclusion. Also at, Lines 306 – 326. No new info is provided. Almost all major metabolic pathways are shown as important and the relative abundances of the genes somewhat corresponds to their relative abundance in the genomes of most microorganisms.

Response: Thank you for your questions on the manuscript, which focuses on the effects of feeding on four different plants on the growth and gut microorganisms of L. migratoria manilensis, comparing the differences in growth indicators and gut bacterial composition of L. migratoria manilensis by feeding on four different plants. Differences in the diversity and abundance of gut microorganisms in these four treatment groups combined with functional prediction analysis, the results discussed in the present study indicated that food plant could affect the gut bacterial structure of L. migratoria manilensis.

  1. Figures: The manuscript currently has 11 figures. I suggest the authors move half of the figures to supplemental and focus on the core figures that best enhance comprehension of their work. Also, key details about how calculations were performed to generate certain figures is missing. For example, how did the authors calculate food utilization efficiency and growth rate etc (Figs 1 & 2)? Table 2 and Figure 3 can be moved to supplemental as an example.

Response: Thank you for your valuable suggestions. For the part of the calculation formula, it is indeed an oversight on our part, and we have added the calculation formula to the article. For the picture part, in order not to bring inconvenience to the reviewers, we will adopt your suggestion later and put some figures into the supplementary material, at present all the figures are still in this article, please understand.

  1. Discussion: As mentioned in the results some of the info here is over-interpreting the data the authors collected. For instance, at Lines 357 to 363 the authors talk about ‘revealing sophisticated symbiotic relationships between gut bacteria and L. migratoria manilensis. The authors do not have any data to support this statement.

Response: We are incredibly grateful to you for pointing out this problem. Line 369-371 have been corrected for the poorly expressed sentence. And for the conclusions section also made changes.

We tried our best to improve the manuscript and made some changes in the manuscript. We appreciate for Reviewers’ warm work earnestly, and hope that correction will meet with approval.

Once again, thank you very much for your comments and suggestions.

Round 2

Reviewer 3 Report

The authors did not make any major changes to the manuscript. The overly high number of figures remain in the main text with none moved to the supplemental as suggested. A few cosmetic changes were made here and there. No additional comments to authors.

Author Response

Dear Reviewer:

Thank you for taking time out of your busy schedule to review and further comments concerning our manuscript entitled “Food sources drive development and variation in Locusta migratoria manliness (Meyen) gut bacterial community” (ID: biology-1882540). Those comments are all valuable and very helpful for revising and improving our paper, as well as the important guiding significance to our research. Now we have carefully corrected the manuscript for this revision, which we hope will be approved by you. Revised portions are marked in red on the paper. The main corrections in the paper and responses to the reviewer’s comments are as follows:

Comments and Suggestions: The authors did not make any major changes to the manuscript. The overly high number of figures remain in the main text with none moved to the supplemental as suggested. A few cosmetic changes were made here and there. No additional comments to authors.

Response: We are incredibly grateful to you for pointing out this problem. We have taken your suggestion to move some of the charts to the supplementary materials. We have also revised the simple summary and abstract sections to better illustrate the topic of the article.

I hope that the changes I’ve made resolve all your concerns about the article. I’m more than happy to make any further changes that will improve the paper and/or facilitate successful publication.

Once again, thank you very much for your comments and suggestions.
